# *BRITTLE CULM17*, a Novel Allele of *TAC4*, Affects the Mechanical Properties of Rice Plants

**DOI:** 10.3390/ijms23105305

**Published:** 2022-05-10

**Authors:** Guangzheng Li, Xiaofang Zeng, Yan Li, Jianrong Li, Xiaozhen Huang, Degang Zhao

**Affiliations:** 1The State Key Laboratory of Green Pesticide and Agricultural Biological Engineering, Center for Research and Development of Fine Chemicals, Guizhou University, Guiyang 550025, China; liyifei2008@163.com; 2The Key Laboratory of Plant Resources Conservation and Germplasm Innovation in Mountainous Region, Ministry of Education, Institute of Agro-Bioengineering, College of Life Sciences, Guizhou University, Guiyang 550025, China; xfzeng1@gzu.edu.cn (X.Z.); yli@gzu.edu.cn (Y.L.); jrli@gzu.edu.cn (J.L.); huangxioazhen012@163.com (X.H.); 3Guizhou Plant Conservation Center, Guizhou Academy of Agricultural Sciences, Guiyang 550006, China

**Keywords:** Guizhou landrace rice, brittle stem mutant, gene mapping, gene functional analysis

## Abstract

Lodging resistance of rice (*Oryza sativa* L.) has always been a hot issue in agricultural production. A brittle stem mutant, *osbc17*, was identified by screening an EMS (Ethylmethane sulfonate) mutant library established in our laboratory. The stem segments and leaves of the mutant were obviously brittle and fragile, with low mechanical strength. Examination of paraffin sections of flag leaf and internode samples indicated that the number of cell layers in mechanical tissue of the mutant was decreased compared with the wild type, Pingtangheinuo, and scanning electron microscopy revealed that the mechanical tissue cell walls of the mutant were thinner. Lignin contents of the internodes of mature-stage rice were significantly lower in the mutant than in the wild type. By the MutMap method, we found candidate gene *OsBC17*, which was located on rice chromosome 2 and had a 2433 bp long coding sequence encoding a protein sequence of 810 amino acid residues with unknown function. According to LC-MS/MS analysis of intermediate products of the lignin synthesis pathway, the accumulation of caffeyl alcohol in the *osbc17* mutant was significantly higher than in Pingtangheinuo. Caffeyl alcohol can be polymerized to the catechyl lignin monomer by laccase ChLAC8; however, ChLAC8 and OsBC17 are not homologous proteins, which suggests that the *osbc17* gene is involved in this process by regulating laccase expression.

## 1. Introduction

Rice (*Oryza sativa* L.) is a globally important food crop that serves as the major food source for more than 50% of the world’s population. In terms of the area planted in rice, China accounts for approximately one-sixth of the total rice planting area in the world (National Bureau of Statistics, http://www.stats.gov.cn/, accessed on 1 February 2022). Rice yield is closely coupled to food security, both in China and worldwide.

Because lodging seriously decreases rice yield and quality, improvement of the mechanical strength of rice stems and enhancement of lodging resistance has long been a desirable characteristic in rice agricultural production. Rice lodging is closely related to various factors, such as plant morphological characteristics, chemical components, epidermal modifications, and other features related to plant growth. According to previous studies, alternation of plant cell wall components is the main reason for rice stem lodging and brittleness.

In eukaryotes, the cell wall is one of the most important characteristics to distinguish animal cells from plant cells. The cell wall is very important for the construction of plant structure, morphology, and function. The cell wall maintains the morphology of cells, enhances the mechanical strength of cells, and also participates in the physiological activities of cells. Differences in the structure and composition of the rice cell wall affect the mechanical strength of the rice stem [1]. During the development of vessels and tracheary elements, the cell wall has a secondary thickening modification process, which plays a supporting role in plants. The main substances of secondary thickening are cellulose and lignin. Cellulose is the basic material of a cell wall. Cellulose microfibrils and hemicellulose cross-link to form the framework of the secondary wall. Lignin is the main material of secondary thickening of the cell wall, which can increase the mechanical strength of the cell wall and the resistance to pathogens [2,3].

Cellulose in the plant cell wall is synthesized by the cellulose synthase complex (CSC), the basic unit of which is cellulose synthase (CESA) [4]. In recent years, researchers have screened and obtained many brittle stem mutants by constructing a rice mutation library, and cloned several genes related to rice cell wall biosynthesis by using continuously improved molecular biology and genetic experimental techniques, most of which are genes that directly or indirectly regulate the rice cellulose synthesis pathway. At present, more than ten regulatory genes related to rice brittle stem have been reported. Among these genes, BC1 gene encodes the COBRA-like protein, which is expressed in thick-walled cells and vascular bundles at the stage of rice development. The mutation of this gene will reduce the thickness of the cell wall and the content of cellulose, and increase the content of lignin, resulting in brittle stems and leaves [5]. BC3 encodes a classical motor protein OsDRP2B, and the mutation of this gene will reduce the cellulose content in leaves and roots [6]. The BC6 gene encodes the OsCesA9 protein, which plays an important role in the deposition of cellulose in the secondary wall [7]. Brittle stem genes BC7 and BC11 are mutant forms of OsCesA4 gene. The whole plant of these two mutants become brittle and the cellulose content decreases [8,9]. The mutation of the OsCesA7 gene in the S1-24 mutant cause brittle stems and leaves, dwarf plants, drooping leaves and stems, reduced tiller number and seed setting rate, decreased cellulose content in stems, and increased hemicellulose content compared with the wild type plants [10]. OsCesA4, OsCesA7, and OsCesA9 are three different catalytic subunits of cellulose synthase, which may form a cellulose synthesis complex and participate in secondary cell wall synthesis [11]. Studies on the GA deletion mutant (d18-AD) and GA response mutant (SLR1) showed that the d18-AD mutant had fewer sclerenchyma cells, the thickness of sclerenchyma cell wall decreased, and the content of cellulose decreased; however, the sclerenchyma cells of the SLR1 mutant were increased, as well as cell wall thickness and the cellulose content. Further studies show that NAC29/31 is the top transcription factor controlling the formation of secondary wall, which can activate MYB61 and then activate the expression of the secondary wall cellulose synthesis gene (CESA). The regulatory pathway NAC29/31-MYB61-CESA affects rice secondary cellulose synthesis and was controlled by the interaction between SLR1 and NAC29/31. The gibberellin (GA) signal can promote cellulose synthesis by inhibiting the interaction between SLR1 and NACs [12]. Moreover, the OsMYB103L gene encodes an MYB transcription factor, which can interact with SLR1 and participate in the regulation of a cellulose synthesis pathway mediated by a GA signaling pathway [13].

The synthesis of a lignin monomer goes through the phenyl-propanoid pathway, starting from the deamination of phenylalanine (or tyrosine) to form cinnamic acid, and finally forms three main monomers (p-coumaryl alcohol, coniferyl alcohol, and sinapyl alcohol) through a series of hydroxylation, methylation, and reduction reactions. The phenyl propanoid pathway is a general and conservative pathway in plant secondary metabolism, which is catalyzed by 14 enzymes, including TAL, PAL, C4H, C3H, F5H, 4CL, CCoAOMT, CCH, CQT, CST, COMT, CCR, CAD, and SAD [14,15]. At present, there are few studies related to the lignin synthesis ability in rice. The reported rice cinnamyl alcohol dehydrogenase deletion mutant fc1 reduces the mechanical strength of the stem, semi-dwarf plant height, 1000 grain weight, and yield per plant. In the mutant, the loss of cinnamyl alcohol dehydrogenase resulted in the decrease in cell wall thickness and lignin content. Further tissue and biochemical analysis showed that p-hydroxyphenyl monomer and o-methoxyphenyl monomer were significantly reduced in the cell wall of the fc1 mutant [16]. Compared with rice, lignin synthesis related genes in *Arabidopsis* and Maize have been more extensively studied in past years. In *Arabidopsis*, more than 30 genes related to lignin monomer synthesis have been cloned and identified, including PAL, C3H, C4H, 4CL, HCT, CCR, CAD, COMT, CCoAOMT, LAC and other key enzyme genes in the lignin metabolism pathway [17,18]. In maize, mutations in genes related to lignin synthesis usually lead to a reddish-brown midrib of leaves, so it is named the brown midrib mutant (bm). Several independent BM gene loci have been found in maize. The bm1 mutant showed decreased activity of cinnamyl alcohol dehydrogenase (CAD); a key enzyme of one-carbon metabolism pathway methylenetetrahydrofolate reductase (MTHFR) inactivated in the bm2 mutant; and cafeate O-methyltransferase (COMT) activity decreased in the bm3 mutant [19,20].

If rice has a strong cell wall, it can significantly improve the lodging resistance; however, the rice plants which have weaker cell walls easily degrade naturally, which is friendly to the environment. The plant cell wall synthesis pathway is a complex and fine regulatory network. At present, there are still many unknown mechanisms for plant cell wall components and material accumulation. There are many genes involved in the synthesis of cell walls. An in-depth understanding of the specific functions of related genes can make it possible to artificially control the synthesis of plant cell walls. To contribute to the elucidation of the major genes responsible for brittle stems and lodging in rice as well as the cell wall construction process, we selected *osbc17*, a brittle stem mutant of Pingtangheinuo, from a rice EMS mutant library previously established by our research team. We used this mutant to locate, clone, identify, and functionally analyze a key gene regulating brittle stems in rice.

## 2. Results

### 2.1. Phenotypes of Wild-Type (WT) Pingtangheinuo and Brittle Stem Mutant osbc17

Pingtangheinuo is a local variety of *japonica* rice collected from Pingtang County, Guizhou Province, China. While screening a Pingtangheinuo EMS mutant library established at an early stage in our laboratory, we identified a brittle stem mutant, designated as *osc17*. After six generations of continuous selfing, a structurally stable mutant was obtained. Compared with the Pingtangheinuo wild type, the *osbc17* brittle stem mutant was significantly dwarfed (Figure 1A). When bent by external forcing, the stems and leaves of *osbc17* were more easily broken than those of Pingtangheinuo. After bending, the wild type was still connected by fibers at the fracture, whereas the mutant was completely severed (Figure 1B,C).

Because leaves and stems of the brittle stem mutant *osbc17* were less resistant to external force bending relative to the Pingtangheinuo WT (Figure 1B,C), we analyzed the degree of mechanical strength reduction in *osbc17* to identify the gene mutation responsible for this change. We randomly selected 20 plants each of Pingtangheinuo, *osbc17*, and their F_1_ progeny at the rice heading stage and measured the tensile strength and bending resistance of their flag leaves and stems. Our analysis revealed that the mechanical strength of *osbc17* was significantly lower than that of the WT, whereas no significant difference in mechanical strength was observed between hybrid and WT plants (Figure 1D). According to previous studies, decreases in cell wall thickness and cellulose contents as well as abnormal cell wall development are directly responsible for the increased brittleness of rice brittle stem mutants (Zhang and Zhou, 2011) [1]. During vegetative and reproductive crop growth, cell wall secondary thickening plays an important role in plant support. The main substances accumulated during this process are cellulose and lignin. In the present study, we therefore measured the cellulose and lignin contents of flag leaves, leaf sheaths, and stems of 20 randomly selected WT and *osbc17* plants at rice maturity. We found that the leaf cellulose content of the brittle stem mutant was significantly lower than that of the WT, whereas cellulose contents of leaf sheaths and stems were not significantly different between the mutant and WT. The lignin contents of leaves, leaf sheaths, and stems of the brittle stem mutant were all lower than those of the WT, and these differences were extremely significant (*p* < 0.01; Figure 1E,F).

### 2.2. Cytological Analysis and Determination of Cell Wall Composition

Mechanical tissue is the main supporting and protective tissue in rice plants (Yang, 2014). Changes in the morphology or number of mechanical tissue cells are mainly responsible for the dwarfed nature and fragility of brittle stem mutants. To determine the specific reason for the decreased mechanical strength of the mutant at the plant tissue level, we prepared paraffin sections of the stems of mature WT Pingtangheinuo and brittle stem mutant *osbc17* plants and compared the morphology of their mechanical tissues under an optical microscope. These histological and cellular observations revealed that the internodes of the mutant had smaller diameters, thinner mechanical tissue epidermal regions, and fewer cell layers (Figure 2A,B,E,F), all of which may explain why mutant internodes had a significantly lower mechanical strength than that of the WT.

The lodging resistance of rice plants is affected not only by decreases in the thickness of mechanical tissue and the number of cell layers, but also by morphological and structural changes to mechanical tissue cell walls. To check for possible changes in mechanical tissue cell walls of the brittle stem mutant *osbc17*, we sampled the flag leaves and stems of the WT and *osbc17* at rice maturity. The samples were fixed with glutaraldehyde fixing solution and then gold plated, and the cell wall structure of their mechanical tissue was observed under a scanning electron microscope. This observation revealed that the cell walls of mechanical tissue cells of flag leaves and stems were significantly thinner in the mutant (Figure 2C,D,G,H), and that cell wall secondary thickening was inhibited. These differences may be the main explanation for why the mechanical strength of the brittle stem mutant plants was significantly lower than that of the WT.

### 2.3. Gene Mutation Mapping

The proportion of individuals with the brittle stem trait in an F_2_ generation produced by reciprocal crossing of *osbc17* and Pingtangheinuo wild type (WT) plants was consistent with a 3:1 segregation ratio (χ^2^ = 0.42 < χ^2^_0.05_ = 3.84), thus indicating that the brittle stem characteristic of the *osbc17* mutant was caused by a pair of recessive alleles. To reveal the exact gene mutation responsible for rice stem brittleness, we used the MutMap method to locate and clone the target gene from F_2_ generation hybrid progeny, which was constructed by crossing the mutant with the wild type, possessing either extremely brittle or non-brittle stems.

To map the chromosomal site of *OsBC17*, 208 SNP loci were obtained by calculating Δ(SNP index), and their annotation results by ANNOVAR software were extracted. Among these SNPs, the SNP loci causing stop loss, stop gain, non-synonymous mutation, and alternative splicing are preferentially selected as candidate mutation sites. By examining data related to the candidate mutation site in the Rice Genome Annotation Project database (http://rice.plantbiology.msu.edu/index.shtml, accessed on 1 April 2022), we localized the mutation of interest to a physical position of 14,682,808 on chromosome 2. The mutation site we found is located at locus LOC_Os02g25230. This point mutation, from guanine (G) to adenine (A) in the exon of the candidate gene, changed the tryptophan-encoding codon TGG to the termination codon TGA, thereby resulting in early termination of candidate gene expression (Appendix A).

Next, we designed PCR primers (forward primer: 5′-gtcccctatcccaatctc-3′; reverse primer: 5′-cacagagttaccacctgc-3′; result as a 571 bp amplified product) to amplify the candidate region according to the gene coding sequence found in the rice genome annotation database. Using these primers, we amplified the candidate gene region from WT and *osbc17* brittle stem mutant plants and analyzed the resulting products by agarose gel electrophoresis (Appendix A). Gel electrophoresis outcomes were recovered for sequencing, which proved the point mutation existed (Appendix A)

### 2.4. Gene Functional Verification

#### 2.4.1. Gene Functional Verification by Gene Knockout

To further confirm the candidate gene as the cause of the brittle stems of the rice mutant, we constructed a knockout vector for this gene using CRISPR/Cas9 technology. The gene knockout vector was genetically transferred into the WT (Pingtangheinuo) by *Agrobacterium tumefaciens*-mediated genetic transformation, and gene function was verified after obtaining transgenic plants.

Plants of two knockout mutants, *cr-2-2* and *cr-2-6*, were significantly dwarfed compared with the Pingtangheinuo WT (Figure 3A) and had more fragile stems and leaves. After fracturing, WT leaves and stems were still connected at the fracture by fibers, whereas those of the mutants were completely severed (Figure 3B,C).

As shown in Figure 3B,C, the tolerance of knockout mutants *cr-2-2* and *cr-2-6* to external force bending was weaker than that of the Pingtangheinuo WT. To quantify the mechanical strength reduction, we randomly selected 20 plants each of T_1_ generation *cr-2-2* and *cr-2-6* plants and Pingtangheinuo plants at rice maturity and measured the tensile resistance of their flag leaves and the bending resistance of their stems. Our measurements confirmed that the mechanical strength of knockout mutants *cr-2-2* and *cr-2-6* was decreased significantly compared with the WT, Pingtangheinuo (Figure 3D). As mentioned above, reductions in various agronomic characters and the mechanical strength of the *osbc17* brittle stem mutant were accompanied by a significant decrease in cell wall lignin contents. To explore whether the cell wall chemical composition of candidate gene knockout mutants *cr-2-2* and *cr-2-6* was different from that of the WT, we measured the cellulose and lignin contents of flag leaves, leaf sheaths, and stems of 20 randomly selected mature WT plants and T_1_ generation *cr-2-2* and *cr-2-6* knockout mutants. Compared with the WT, no significant difference was detected in the cellulose contents of leaves, leaf sheaths, and stems of knockout mutants *cr-2-2* and *cr-2-6*, whereas the lignin contents of these organs were significantly lower in *cr-2-2* and *cr-2-6* (Figure 3E,F).

To determine the reason for the decreased mechanical strength of the knockout mutants at the cellular level, we made paraffin sections of mature WT and knockout mutant *cr-2-2* and *cr-2-6* plants and looked for morphological differences in their mechanical tissues under an optical microscope. Our observations revealed that the number of cell layers in the stem epidermis (i.e., mechanical tissue) of the knockout mutants was lower than that of the WT (Figure 4A–C).

We examined mechanical tissue between the flag leaf and stems of plants of the WT Pingtangheinuo and knockout mutants *cr-2-2* and *cr-2-6* by scanning electron microscopy. Our observations revealed thinning of the cell walls of mechanical tissue cells located between the flag leaf and internodes of the knockout mutants (Figure 4D–I). This result suggests that cell wall secondary thickening was blocked in the knockout mutants *cr-2-2* and *cr-2-6*. This mechanical tissue cell wall thinning was similar to that observed in the brittle stem mutant *osbc17*. In terms of agronomic characteristics, plant mechanical strength, and cell morphology, the knockout mutants *cr-2-2* and *cr-2-6* were therefore significantly different from WT Pingtangheinuo but similar to the brittle stem mutant *osbc17*. These observations suggest that the mutation of the candidate gene is the main reason for the reduction in the stem mechanical strength of the brittle stem mutant *osbc17*.

#### 2.4.2. Gene Functional Verification by Overexpression of the *OsBC17* Gene in the Brittle Stem Mutant

A plasmid carrying an overexpression vector of the candidate gene, *OsBC17* (Appendix A), was introduced via *A. tumefaciens* to transform *osbc17* mutant rice. Given the possibility of false positives or transient transformation, we screened GUS-positive seedlings for the presence of the *OsBC17* gene by PCR (Appendix A) and identified 12 transgenic seedlings. When the stems and leaves of these transgenic plants were subjected to bending under the action of external force, the fracture points were not visually different from those of the Pingtangheinuo WT (Figure 5A,B).

To further compare the mechanical strength of the transgenic plants and the Pingtangheinuo WT, we randomly selected 20 T_1_ generation transgenic plants and 20 Pingtangheinuo WT plants at the rice maturity stage and measured the tensile and bending resistance of their flag leaves and stems. We found that the mechanical strength of the transgenic plants was not significantly different from that of Pingtangheinuo (Figure 5C).

When we examined paraffin sections of internode tissues and cells, we detected no significant difference in the number of epidermal cell layers (i.e., mechanical tissue) of stem segments of transgenic and WT plants (Figure 5D,E).

Finally, scanning electron microscopic observations of mechanical tissue cell walls of stems of Pingtangheinuo WT and transgenic plants revealed no major differences between them (Figure 5F–I). Overall, the transgenic plants were not substantially different from the WT in terms of agronomic characteristics, plant mechanical strength, or cell morphology, thus demonstrating that restoration of the candidate gene, *OsBC17*, was able to increase the stem mechanical strength of the brittle stem mutant *osbc17*.

### 2.5. Lignin Metabolomics Analysis

Our measurements of cellulose and lignin contents of flag leaf and stem cell walls of WT Pingtangheinuo and brittle stem mutant osbc17 plants clearly revealed that a decrease in cell wall lignin contents was closely related to the thinning of mechanical tissue cell walls and decreased plant mechanical strength. The synthesis of the lignin monomer goes through the phenyl propanoid pathway, which is highly conserved in plant secondary metabolism, has been well characterized, and the genes coding for the various steps of lignin biosynthesis have been discovered (Raes et al., 2003) [17]. The research on determining levels of variation among aspen (*Populus tremuloides* Michx.) populations by using SNPs as molecular markers showed that the gene locus involved downstream of the lignin biosynthesis pathway evolved faster than those genes involved upstream (Kelleher et al., 2012) [21]. Thus, SNP variability might be larger downstream of the lignin biosynthesis pathway, which we expected the SNPs we screened were mainly involved in.

To determine why the cell wall lignin content of the brittle stem mutant was lower than that of the WT, we subjected samples of the WT and the brittle stem mutant to lignin metabolomics analysis. Metabolomics analysis is mainly used to detect and screen biologically important, significantly differentially accumulated metabolites in biological samples and to clarify the metabolic processes and change mechanisms of organisms on this basis.

As shown in Figure 6, the only metabolite in the lignin synthesis pathway that was up-regulated in the brittle stem mutant *osbc17* relative to the Pingtangheinuo WT was caffeyl alcohol, with a significant fold change of 2.84, whereas no significant changes were noticed among other metabolites. The fact that the cell wall lignin content of the *osbc17* mutant decreased while the level of caffeyl alcohol increased suggests that the metabolic pathway leading from caffeyl alcohol to other lignin precursors was blocked in the *osbc17* mutant. Although caffeyl alcohol is not one of the three common lignin alcohol monomers (i.e., p-coumaryl, coniferyl, and sinapyl alcohols), a substrate-specific laccase, ChLAC8, has been found in the seed coat of *Cleome hassleriana* that oxidizes caffeyl alcohol and sinapyl alcohol, but not coniferyl alcohol [22]. This laccase can catalyze the polymerization of caffeyl alcohol to catechyl lignin. Because ChLAC8 and OsBC17 are not homologous proteins (Appendix A), in rice, the *OsBC17* gene may regulate expression of laccase, which is ChLAC8′s homologous protein.

### 2.6. Differential Gene Expression between the osbc17 Mutant and WT

To investigate the biological function of OsBC17, RNA sequencing analysis was conducted to compare the transcriptome profiles of differentially expressed genes (DEGs) in the leaves of 4-week-old seedlings of the Pingtangheinuo wild type and osbc17 mutant. A total of 4177 DEGs were identified between the osbc17 mutant and wild type, of which 2345 genes were up-regulated, and 1832 genes were down-regulated (Figure 7A).

All identified DEGs were clustered according to their biological functions. Gene ontology (GO) annotation and KEGG pathway analysis were performed at the same time. The GO functional annotation was further divided into three major categories: biological process, cellular component, and molecular function. There are subcategories at various levels under each major category. According to the GO classification chart, in the biological process, differential genes are mainly involved in the cellular process and metabolic process; in cell components, differential genes are mainly involved in membrane, membrane part, organelle, and cell; in molecular function, differential genes are mainly involved in binding and catalytic activity (Figure 7B).

According to the KEGG annotation and classification results, the differential genes involved in the KEGG metabolic pathway were divided into five branches: cellular processes, environmental information processing, genetic information processing, metabolism, and organic systems. It can be seen from the KEGG pathway classification chart that the differential genes are mainly involved in the metabolism pathway (Figure 7C).

Through the analysis of lignin metabolomics, it was found that the decrease in lignin content in the cell wall of brittle stem mutant osbc17 may be due to the obstruction of the metabolic pathway of caffeyl alcohol to C-lignin precursors. Laccase is the main catalytic enzyme in the metabolic pathway of lignin monomer biosynthesis. In order to further study the effect of the osbc17 mutation on lignin synthesis, we analyzed the expression of genes related to laccase in the wild type and the osbc17 mutant. We found 18 putative laccase precursor genes among the differential genes screened by RNA sequencing. Thirteen of those genes’ expression was significantly decreased in the osbc17 mutant. By calculating the value of log2 (mutant FPKM/wild type FPKM) to show the level of gene expression difference between mutant and wild type, the most significant expression decline among those genes is LOC_Os11g42200, reaching −10.27 (Figure 8).

## 3. Discussion

Previous studies investigating the mechanical strength reduction in rice brittle stem mutants have mainly focused on the direct or indirect obstruction of the cellulose synthesis pathway. The *osbc17* brittle stem mutant examined in the present study was not significantly different from the WT in regard to flag leaf and stem cellulose content, but its lignin content was significantly reduced. The new *osbc17* brittle stem mutant thus has unique characteristics. To identify the allele responsible for the brittle stem characteristic of the *osbc17* mutant, we used the MutMap method to locate candidate SNP sites. Thus, we obtained the candidate gene loci LOC_Os02g25230. Thus far, in global databases included in the Rice Genome Annotation Project (http://rice.uga.edu, accessed on 1 April 2022), the gene is annotated as ‘conserved gene of unknown function, expressed protein’. To functionally verify the candidate gene, we constructed knockout mutants using CRISPR-Cas9 technology. The generated knockout mutants were similar to the brittle stem mutant in several respects: their mechanical strength was significantly reduced compared with the WT, their mechanical tissue cell walls were thinner, and their cell wall lignin contents were significantly lower. According to these results, the candidate gene, *OsBC17*, is responsible for the brittle stem trait. *OsBC17* is a novel allele of *TAC4*, which can regulate rice shoot gravitropism by increasing the indole acetic acid content and affecting the auxin distribution, and significantly increase the tiller angle of *TAC4* deficiency mutation [23]. OsBC17 protein, the expression product of the *OsBC17* gene, does not have any known conserved domain. The homologous proteins found from the NCBI website are all proteins with unknown function, so it seems impossible to predict the biological function of the OsBC17 protein. Our lignin metabolomics analysis revealed that the lignin content of the *osbc17* brittle stem mutant was decreased, whereas its caffeyl alcohol content, related to the lignin synthesis pathway, was increased compared with the WT. This result suggests that the metabolic pathway in which lignin precursors are synthesized via the polymerization of caffeyl alcohol is blocked in the *osbc17* mutant. The polymerization of caffeyl alcohol to catechyl lignin is an uncommon lignin synthesis pathway that has not been previously reported in rice. ChLAC8 found by Wang et al. [22], as well as OsBC17, are not homologous proteins, thus the *Osbc17* gene may regulate expression of laccase, which is ChLAC8’s homologous protein. To investigate the biological function of OsBC17, there were 13 putative laccase precursor genes, the expression levels of which were significantly decreased in the *osbc17* mutant, found through RNA sequencing. The correlation between the *OsBC17* gene and putative laccase precursor genes remains to be further studied.

At present, there are few reports that focus on the synthesis of lignin catalyzed by laccase in rice and the regulation of laccase gene expression, and further study of the new *osbc17* brittle stem mutant should help elucidate this whole process. Thus, the principle of the secondary thickening process of the cell wall can be improved. Furthermore, in-depth study on the regulation of laccase gene expression will help to provide a new direction for the study of lodging resistance and disease resistance in rice.

## 4. Materials and Methods

Pingtangheinuo, the wild type japonica rice variety used in this study, was collected from Pingtang County, Guizhou Province, China. The brittle stem mutant, *osbc17*, is a genetically stable brittle stem mutant screened from a mutant library constructed by treating Pingtangheinuo with the chemical mutagen EMS (Ethylmethane sulfonate). All plant materials in this study were collected and preserved by the Key Laboratory of Plant Resources, Conservation, and Germplasm Innovation of Mountainous Regions, Guizhou University, Guizhou, China.

### 4.1. Rice Hybridization

Rice hybridization was carried out according to the method of Yu [24] with slight modification. The brittle stem mutant *osbc17* was hybridized with the Pingtangheinuo WT, and the resulting F_1_ generation was self-crossed to yield the F_2_ generation used to construct a fine mapping population.

### 4.2. Measurement of Physical Properties

Plant physical properties were measured according to the methods of Zhang et al. [25]. Flag leaves and stems of the Pingtangheinuo WT and the brittle stem mutant *osbc17* were collected from the field at the full heading stage and protected from desiccation until testing. To determine tensile strength, lay each sample flat on a test board, with one end glued to the test bench and the other end attached to a portable electronic device. Slowly apply tension by pulling until the sample breaks.

### 4.3. Cytological Analysis

Paraffin sectioning and scanning electron microscopic observation followed the methods of Cheng et al. [26] and Cao et al. [27], respectively, with slight modifications. To examine the morphology of rice stem tissues and cells, we used the stems of Pingtangheinuo and *osbc17* mutant plants sampled at the rice heading stage. Paraffin sectioning was carried out to examine the cell size and organization of rice stems. Plant material was cut into small pieces and then put into FAA(Formaldehyde-acetic acid-ethanol Fixative) solution at room temperature for more than 24 h, and then dehydrate it with alcohol gradient. After embedding, the cooled and hardened wax block wrapped with the sample is sliced, stained and observerd under microscope. In order to better understand the changes of cell wall structure between wild type and brittle stem mutant, rice stem tissue was cut into thin slices, then the samples were fixed with 2.5% glutaraldehyde stationary solution at room temperature for more than 24 h. Samples after lyophilization and isoamyl acetate treatment was scaned under scanning electron microscope and take photos.

### 4.4. Determination of Cell Wall Cellulose and Lignin Contents

Cellulose and lignin contents of cell walls of the WT and the brittle stem mutant were measured according to Jin et al. [28]. The rice sample was ground into fine powder in liquid nitrogen, and then 35 μL of 0.01% sodium azide (NaN_3_), 35 μL Amylase solution, 17 μL branched chain amylase solution was used to remove starch. Then fully dried to obtain isolated cell wall samples. The cellulose in the cell wall sample was digested and decomposed into glucose, and its absorbance value at 625 nm was measured. Draw the standard curve according to the absorbance value reading of glucose concentration standard solution, and calculate the content of glucose (i.e., crystalline cellulose) by comparing the absorbance of the sample with the standard curve. Similarly, the lignin in the cell wall sample was digested and decomposed, and the absorbance of the decomposed solution at 280 nm was measured. Draw the standard curve according to the absorbance value reading of the standard solution, and calculate the lignin content by comparing the absorbance of the sample with the standard curve. All the data were analyzed using Excel 2010 and SPSS 26.0. The data were analyzed using Excel 2010 and SPSS 26.0.

### 4.5. Genomic DNA Extraction

Rice sample DNA extraction was carried out according to Chen et al. [29], with slight modification. About 1 g of fresh rice leaves were cut into small pieces and then quickly grinded to powder in liquid nitrogen. Quickly add the powder into the centrifugal tube containing CTAB extraction buffer, mixed thoroughly and keep the centrifugal tube in a 65 °C water bath for 30 min and then centrifugation at 12,000 rpm for 10 min. The crude extraction after centrifugation is dissolved in chloroform and isoamyl alcohol mixture. NaCl solution with final concentration of 0.2~0.4 M was added to the supernatant, then absolute ethanol equivalent to twice the total volume of the liquid was added to precipitate DNA.

### 4.6. Gene Mapping by MutMap Method

MutMap method was carried out to map candidate gene according to previous studies [30,31,32,33]. The whole genome DNA extracted from the *osbc17* mutant and WT parents, 20 non-brittle plants and 20 brittle plants from the F_2_ population, were randomly interrupted to 350 bp fragments by Covaris Crusher. TruSeq Library Construction Kit was used to build the library, and the whole library was prepared by DNA terminal repair, adding ployA tail, adding sequencing adapter, purification, and PCR amplification. Then, the library was sequenced using an illumina HiSeqTM PE150 platform. The original image data file obtained by sequencing was transformed into the original sequenced reads by Casava Base Calling analysis, also known as raw data or raw reads. Because raw reads may contain low-quality data such as sequencing adapter, they need to be filtered to obtain clean reads that can be used for subsequent analysis. A total of 60.435 Gb of clean reads were obtained that were aligned to the Nipponbare genome, resulting in identification of 1151428 SNP (single nucleotide polymorphism) positions.

The parent WT was selected as the reference parent. For each SNP position, the value of the SNP index (the ratio of clean reads harboring SNPs compared with the reference) was determined, and a graph relating SNP positions and the SNP index was generated for all 12 rice chromosomes. For the brittle stem progeny, the SNP index related to the brittle stem phenotype should be close to 1; on the contrary, the SNP index related to brittle stem phenotype should be close to 0 for non-brittle stem progeny. In order to avoid the impact of sequencing and/or alignment errors, the sites with SNP index less than 0.3 and SNP number less than 7 in both progeny and the sites with SNP deletion in one progeny were filtered out. After filtering, 2469 SNP marker loci were obtained.

In order to obtain precise gene loci, the difference in SNP index between the two progeny, namely Δ(SNP index), is calculated:Δ(SNP index) = SNP index (brittle stem) − SNP index (non-brittle-stem)

Conduct 1000 replacement tests, select 95% confidence level (green line) as the screening threshold, and the window greater than the threshold as the candidate interval, according to the previously described methods [34,35,36,37,38,39] (Appendix A).

### 4.7. Polymerase Chain Reaction

The coding sequence (CDS) of the candidate gene was found from the Rice Genome Annotation Project (http://rice.plantbiology.msu.edu/index.shtml, accessed on 1 April 2022), by which according CDS the PCR amplification primers of the candidate gene were designed. eTaq 5 μL, both forward primer and reverse primer 0.5 μL, template DNA 1 μL and ddH2O 3 μL were mixed as PCR reaction system. Reaction conditions: 95 °C 3 min, (95 °C 30 s, 55 °C 30 s, 72 °C 40 s) × 35 cycle, 72 °C 10 min, 4 °C forever.

### 4.8. Gene Knockout by CRISPR/Cas9

Following the methof of Li et al. [40] and Debladis et al. [41], sgRNA sequences suitable for site-directed mutation of candidate genes in rice were designed according to the target design requirements of the CRISPR/Cas9 system using an online target site design database (http://crispr.dbcls.jp, accessed on 20 January 2021). An exonic region in the candidate gene consisting of 30 to 49 bp after the start codon (ATG) was selected as a target for editing (sgRNA sequence: AGCGTCGGCTCCGGCGCCGG). BLAST screening based on this target sequence uncovered no potential off-target sequences. The sgRNA sequence of the candidate gene was then connected to a BGK30 CRISPR/CAS vector to yield a CRISPR/Cas9 sgRNA expression vector driven by the rice U6 promoter, which was suitable for site modification of the candidate gene. After CRISPR/Cas9 editing, plants were screened on hygromycin medium, and 10 transgenic plants were identified on the basis of their hygromycin resistance. Genomic DNA of the 10 plants was extracted and then subjected to PCR amplification using the target-specific sequencing primers (forward primer: 5′-TCTCTCTTGTGGTCGTTGCC-3′; reverse primer: 5′-GGTTGGAGGCGGAGAAGAC-3′).

### 4.9. Lignin Metabolomics Analysis

As described by Fraga et al. [42], the multiple reaction monitoring (MRM) mode of triple quadrupole mass spectrometry is useful for quantitative analysis of metabolites. In MRM mode, the four-stage rod first screens the precursor ions (parent ions) of the target substance and excludes the ions corresponding to other molecular-weight substances to preliminarily eliminate the interference. After the precursor ions are induced and ionized by the collision chamber, they break to form many fragment ions. The fragment ions are filtered through the triple four-stage rod to select a required characteristic fragment ion to eliminate the interference of non-target ions, thereby improving the accuracy of the quantification and enhancing repeatability. After obtaining the mass spectral data of metabolites in different samples, the mass spectral peaks of all substances are integrated according to their peak area, and the mass spectral peaks of the same metabolite in different samples are integrated and corrected. The accumulation of varieties of metabolites in lignin synthesis pathway were detected by liquid chromatography tandem mass spectrometry (LC-MS/MS), following the method of Chen et al. [43]; and Chen et al. [44]. After calculating the fold change (expression ratio between samples) for each metabolite and the corresponding *p*-value (significance) by the Wilcoxon rank sum test, we selected those metabolites with fold change ≥2 or ≤0.5 as significantly differentially accumulated metabolites. A schematic diagram was made of the lignin synthesis pathway, including metabolites differentially accumulated between the WT Pingtangheinuo and the brittle stem mutant *osbc17*.

### 4.10. RNA-Sequencing Analysis

Total RNA was extracted from leaves of 4-week-old mutant and WT plants using an RNA extraction kit (Omega Bio-Tek, Doraville, GA, USA) in accordance with the manufacturer’s instructions. The cDNA libraries were constructed and sequenced on a BGISEQ-500 platform by the Beijing Genomics Institute (Shenzhen, China; http://www.genomics.org.cn, accessed on 30 June 2021). Three biological replicates were prepared for each sequencing library. The DEGseq method was used to screen genes differentially expressed between groups. The differentially expressed genes (DEGs) were selected according to a previously described method [45]. Clusters of orthologous groups (COG) functional classification, gene ontology (GO), and Kyoto Encyclopedia of Genes and Genomes (KEGG) metabolic pathways annotation and enrichment analyses were conducted by accessing the NCBI COG, GO, and KEGG databases, respectively [46,47,48].

## Figures and Tables

**Figure 1 ijms-23-05305-f001:**
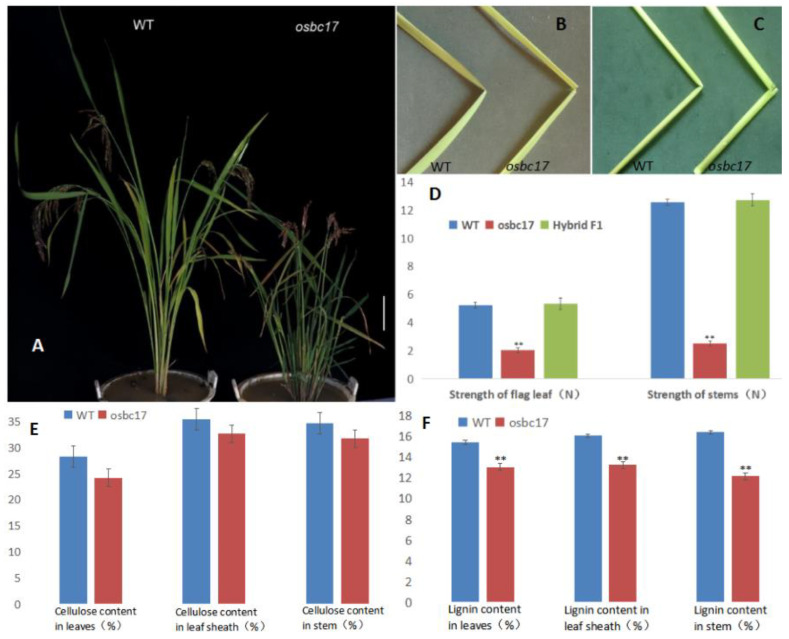
Phenotypes of wild-type Pingtangheinuo and brittle stem mutant *osbc17*. (**A**) Rice plants of wild-type and brittle stem mutant. Bar = 20 cm. (**B**,**C**) Fracturing of wild-type Pingtangheinuo and brittle stem mutant *osbc17* in response to external force. Fracture points of flag leaves (**B**) and stem internodes (**C**) of wild-type and mutant plants. (**D**) Comparison of the mechanical strength of the WT, the *osbc17* mutant, and their F_1_ progeny. **, significantly different at *p* < 0.01. (**E**,**F**) Cellulose contents (**E**) and lignin contents (**F**) of wild-type Pingtangheinuo and brittle stem mutant *osbc17* plants. Asterisks indicate significant differences (**, *p* < 0.01).

**Figure 2 ijms-23-05305-f002:**
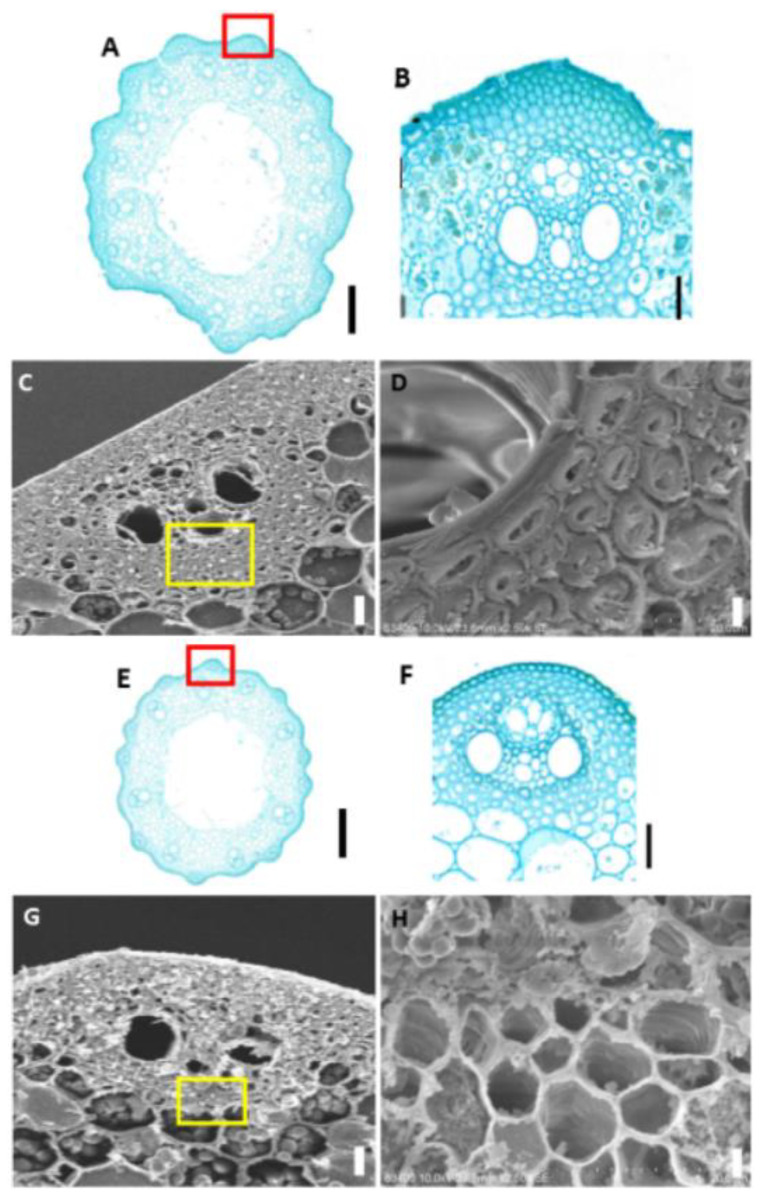
(**A**,**B**,**E**,**F**) Paraffin sections of stems of the Pingtangheinuo wild type (WT) and *osbc17* brittle stem mutant. (**A**,**E**) Transverse sections of whole stem segments of the WT (**A**) and the brittle stem mutant (**E**). The red boxes in (**A**,**E**) are enlarged to show (**B**,**F**), respcetively. (**B**,**F**) Local enlargements of epidermis of the WT (**B**) and brittle stem mutant (**F**). Bar = 1 mm (**A**,**E**) or 20 µm (**B**,**F**). (**C**,**D**,**G**,**H**) Scanning electron microscopic images of wild-type (WT) Pingtangheinuo and brittle stem mutant *osbc17* internodes. (**C**,**G**) Transverse sections of stems of the WT (**C**) and the brittle stem mutant (**G**). The yellow boxes in (**C**,**G**) are enlarged to show (**D,H**), respcetively.(**D,H**) Local enlargements of stems of the WT (**D**) and the brittle stem mutant (**H**). Bar = 20 µm (**C**,**G**) or 4 µm (**D**,**H**).

**Figure 3 ijms-23-05305-f003:**
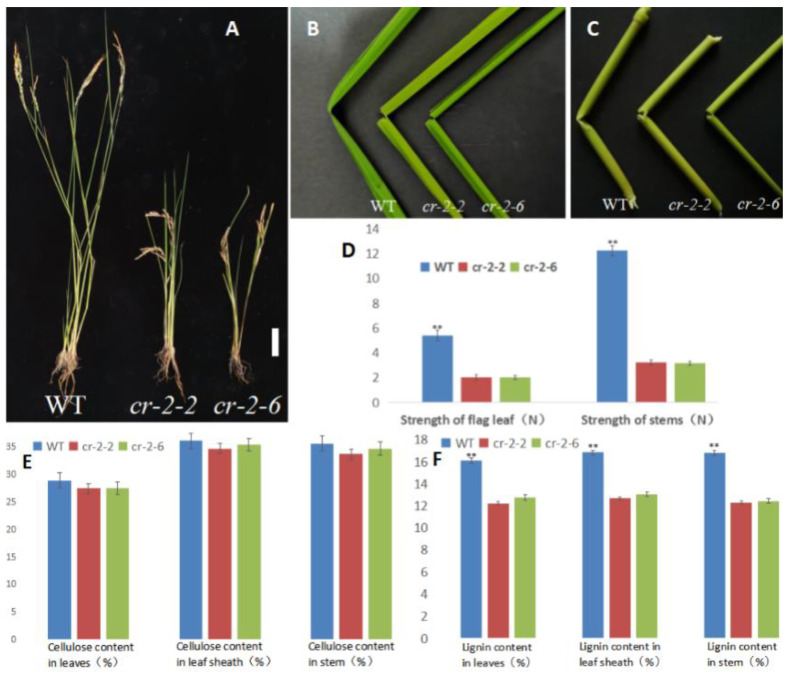
Phenotypes of wild-type and candidate gene knockout rice plants. (**A**) Wild-type, mutant *cr-2-2*, and mutant *cr-2-6* plants. Bar = 10 cm; (**B**,**C**) fracture points of flag leaves (**B**) and internodes (**C**) of wild-type, mutant *cr-2-2*, and mutant *cr-2-6* plants. (**D**) Comparison of the mechanical strength of WT, mutant *cr-2-2*, and mutant *cr-2-6* plants. **, significantly different at *p* < 0.01. (**E**,**F**) Cellulose contents (**E**) and lignin contents (**F**) of wild-type Pingtangheinuo and mutant *cr-2-2* and *cr-2-6* plants. **, significantly different at *p* < 0.01.

**Figure 4 ijms-23-05305-f004:**
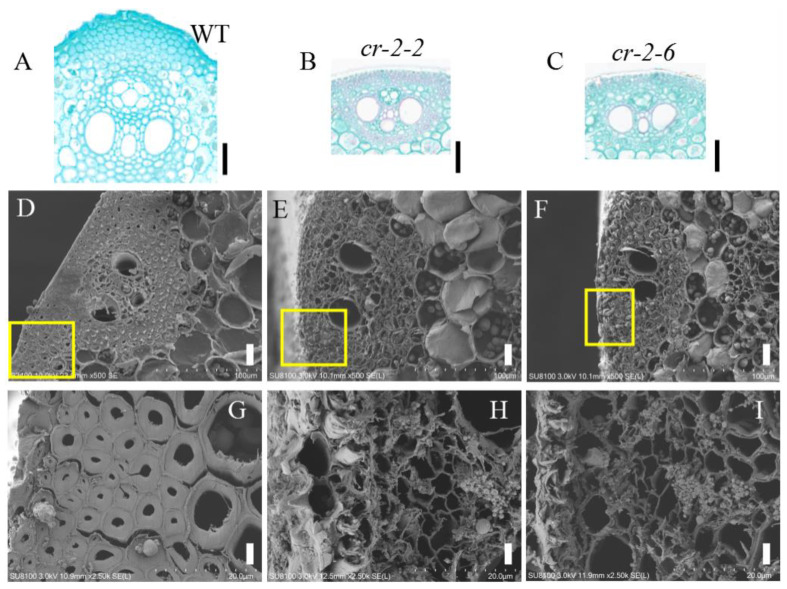
(**A**–**C**) Enlarged images of paraffin sections of stem epidermis of wild-type Pingtangheinuo and mutant *cr-2-2* and *cr-2-6* plants. Bar = 50 µm. Scanning electron microscopic images of internode cross sections of wild-type Pingtangheinuo and mutant *cr-2-2* and *cr-2-6* plants. (**D**–**F**) Transverse sections of wild-type (**D**), *cr-2-2* (**E**), and *cr-2-6* (**F**) internodes (bar = 20 µm). The yellow boxes in (**D**–**F**) are enlarged to show (**G**–**I**), respcetively. (**G**–**I**) Local enlargements of mechanical tissue of wild-type (**G**), *cr-2-2* (**H**), and *cr-2-6* (**I**) internodes (bar = 4 µm).

**Figure 5 ijms-23-05305-f005:**
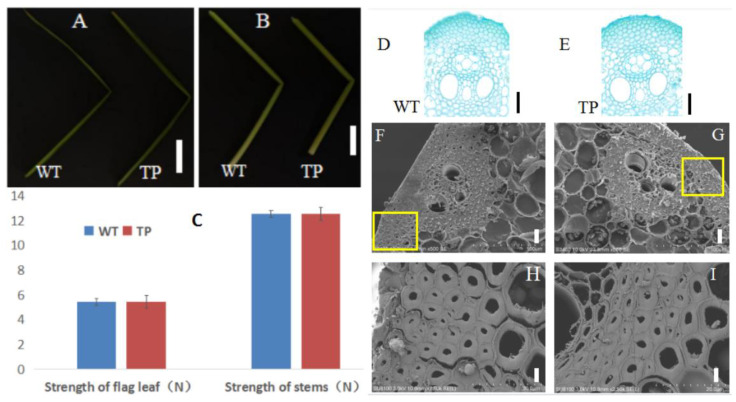
(**A**,**B**) Fractured leaves and internodes of wild-type (WT) and *OsBC17*-overexpressing transgenic (TP) plants. (**A**) Flag leaves (bar = 10 cm); (**B**) internodes (bar = 2 cm). (**C**) Comparison of the mechanical strength of WT and *OsBC17*-overexpressing transgenic plants. (**D**,**E**) Paraffin sections of stem epidermis of Pingtangheinuo wild type (WT) and *OsBC17*-overexpressing transgenic (TP) plants. Bar = 50 µm. (**F**–**I**) Scanning electron microscopic images of stem cross sections of Pingtangheinuo wild type (WT) and *OsBC17*-overexpressing transgenic (TP) plants. (**F**,**G**) Transverse sections of WT (**F**) and TP (**G**) stems (bar = 20 µm); enlarged images of WT (**H**) and TP (**I**) mechanical tissue (bar = 4 µm). The yellow boxes in (**F**,**G**) are enlarged to show (**H**,**I**), respcetively.

**Figure 6 ijms-23-05305-f006:**
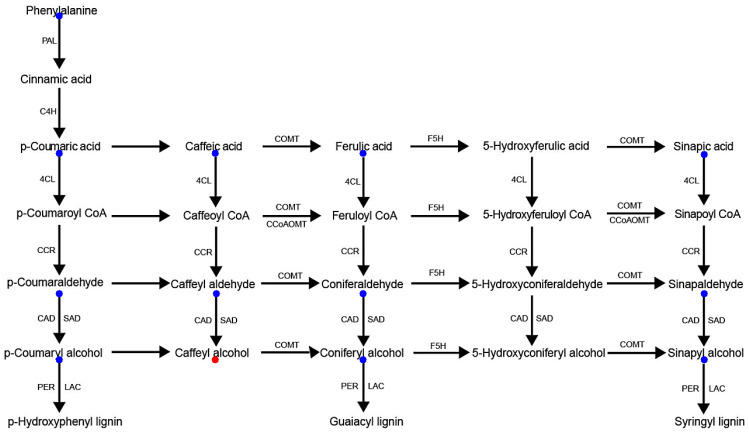
Lignin synthesis pathway-related metabolites differentially accumulated between wild-type Pingtangheinuo rice and brittle stem mutant *osbc17*. Metabolites that were not significantly changed or significantly up-regulated in the brittle stem mutant are indicated by blue and red, respectively.

**Figure 7 ijms-23-05305-f007:**
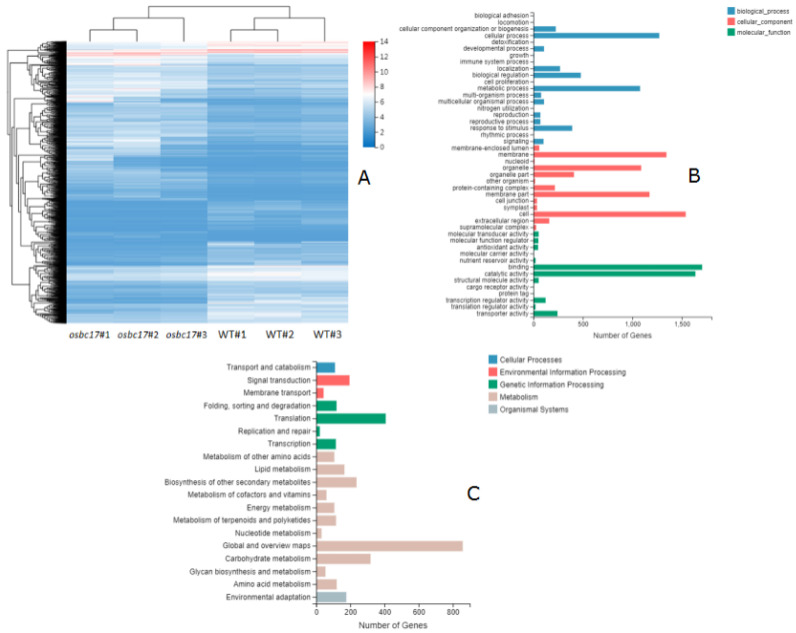
(**A**) Clustering heat map of differential gene expression. (**B**) GO classification of differential genes. Transverse axis: number of genes; longitudinal axis: Gene Ontology. (**C**) KEGG pathway classification of differential genes. Transverse axis: number of genes; longitudinal axis: KEGG pathway.

**Figure 8 ijms-23-05305-f008:**
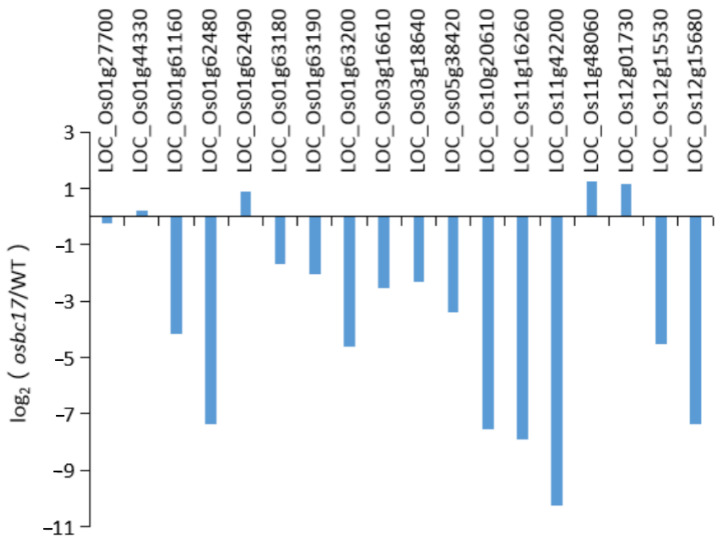
Relative expression difference in putative laccase precursors in wild type and mutant osbc17. Transverse axis: putative laccase precursor gene locus; longitudinal axis: relative gene expression.

## Data Availability

Not applicable.

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
