# Peer review of "BRITTLE CULM17, a Novel Allele of TAC4, Affects the Mechanical Properties of Rice Plants"

_ijms, 2022, doi:10.3390/ijms23105305_

Round 1
Reviewer 1 Report
The work by Li et al. studied lodging resistance in rice plants by combining a candidate gene mapping approach with expression analysis of BRITTLE CULM17 allelic variation at TAC4 lignin-related gene. They work is well framed and rigorously addressed as to justify publication at Int J Mol Sci. However, as part of a major revision recommendation, I bring the following suggestions to the authors.
First, Li et al. should list explicit research goals / hypotheses within the abstract and at the end of the introduction section (last paragraph). This will allow readers focusing on explicit questions addressed by the work. Also related with this point, the introduction still reads to vague and requires more details concerning the nature of lodging resistance, the utility of the candidate gene approach, and the details of the lignin biosynthesis pathway in other plant species beyond rice. Please frame the introduction from the major research gap of the discipline (still lacking in the first introductory paragraph) to concrete research objectives as part of this study (last introductory paragraph).
Second, as part of the analysis depicted in figure 5, authors should also utilize more explicit mixed linear models for the mapping, better at controlling spurious effects than the Δ SNP index. To accomplish this more robust reconstruction of the genomic architecture for abiotic stress tolerance, please refer to figure 3 in Front Plant Sci 2018 9:128, and figures 2-5 in Front Genet 2019 10:954.
Third, when reconstructing the lignin biosynthesis pathway in figure 17, authors should acknowledge that is highly conserved in plant species, and therefore learning from other genus are highly valuable. Concretely, the lignin biosynthesis pathway was first study in poplar, a model tree species, but none reference recognizes this aspect. I recommend authors to explicitly refer to the report Tree Genet Genome 2012 8:821-829, by discussing that SNP variability (for instance in terms of figure 5’s Δ SNP index) is more likely found in downstream genes within the lignin biosynthesis pathway.
Fourth, the discussion (L329) still reads too vague and needs more in depth analysis. Explicitly, please comment, as a recommendation for other studies, how the candidate gene approach is still valid despite overwhelming genomic resources for plant species. In order to reconstruct the utility and validity of the candidate gene approach when addressing the genetic basis of abiotic stresses in plants, please refer chronologically to the intrinsic evolution of the candidate gene approach case reports BMC Genet 2012 13:58, Theor Appl Genet 2012 125(5):1069-85, and Plant Sci 2016 242:250-9. Make sure to close the discussion section with a perspectives paragraph.
Last but not least, the manuscript overall has too many main figures. In order to assist readability, please condense some as subfigures within the same figure panel.
Author Response
Dear Reviewer,
,I have carefully revised my article based on the comments of you. Here I will response each reviewer point raised.
Firstly, we rewrite the introduction, add more detailed research on the synthesis of cell wall components in rice and other plants, and add specific research objectives. Hoping to better explain our theme and goal.
Second, it may not be clearly written in the early version of this article. The SNPs that we screened is not directly obtained by comparing the wild type genome with the mutant genome; but by hybridizing the wild type with the mutant to obtain the F1 generation, and then self crossing the F1 generation to produce the F2 generation. The genetic characteristics of brittle stem mutant traits are analyzed through the separation of F2 generation traits. 20 plants with extremely brittle stem and 20 plants with non-brittle stem were selected from F2 population, and the parents’ and progeny's DNA was extracted to construct gene library. Because the genetic background of the progeny library is exactly the same, the molecular markers showing polymorphism of comparison between brittle stem and non-brittle stem progeny library are likely to be caused by mutation. This method is called MutMap method, which is different from the genotyping by sequencing (GBS) method used in the literature you recommend. Therefore, the calculation method of data is also different. We revised 2.3. Gene mutation mapping hope this question can be explained clearly.
Third, as mentioned in the second point, the SNP sites we screened were not obtained from the genomic comparison between wild-type and mutant. The candidate SNP we obtained is the mutation sites with the greatest difference between the two progeny with extreme brittle and non-brittle phenotype. As for the expression of downstream genes within lignin biosynthesis pathway, we explained in the analysis of gene expression and screened out the hypothetical laccase precursor genes with down-regulated expression.
Forth, the discussion part is modified and supplemented, hoping to meet your requirements.
At last, the charts in this article have been re-organized, hoping to show it more clearly.
Best wishes,
Guangzheng Li and Degang Zhao
Guizhou university
Reviewer 2 Report
Li et al have identified a novel allele of the rice TAC4 gene by mapping and have also functionally validated the observed phenotype by creating CRISPR knockouts. The approach taken to validate the causal mutation is very clear and results clearly support their claim. Following are a few comments which may help increase the manuscript's overall quality:
- Can authors annotate the candidate gene to their molecular function? Since its novel allele of TAC4 which is known for causing increased tiller angle, did authors notice any difference in the tiller angle? Does the novel allele also show the already known function of the genes? Can authors include the information about what position of CDS and the amino acid was changed to stop codon with respect to the gene and protein?
- Can authors better describe the lodging resistant rice mutants and their causal genes or pathway, if known, in the introduction section? Also, adding information about the TAC4 protein, rice tac4 mutant, and its phenotypes could be valuable for readers.
- Authors are strongly suggested to re-organize the figures and tables. Many of the figures can be combined, primer information can go to the supplementary files. For example, figure 6 and 7 and Table 3 can be combined and Figure 8, table 4, table 5 data can be combined. Figure 9 could be a good example of how different figures are arranged as panels.
- Authors must improve the representation of editing results; it should be clearly represented by using some alignment software keeping the PAM of all the lines in one place.
- Differences in the cellulose and lignin contents can be better visualized if they are in a bar chart. Is there any specific reason, why authors have observed more deviation in cellulose content than the lignin content? The large deviation is making the cellulose differences non-significant even though sometimes they look different.
- Authors should clearly describe the methods used for all the experiments. There is no mention of genomic DNA extraction, PCR, Sanger sequencing, vector construction in the method section. Also, the method-related information should be moved to the method section (gRNA and gene editing result section).
Author Response
Dear Reviewer,
I have carefully revised my article based on the comments of you. Here I will response each reviewer point raised.
First, the mutation site we found is located at locus LOC_ Os02g25230. So far, in global databases includes the Rice Genome Annotation Project (http://rice.uga.edu), the gene is annotated as ‘conserved gene of unknown function, expressed protein’. Its allele TAC4 is Os02g045000 at the Rice Annotation Project Database (rapdb. DNA. AFFRC. Go. JP) and annotated as ‘Regulation of tiller angle’. . The position changes of CDs and amino acids have been given in the modified article. We did observe a change in the tillering angle of the mutant.
Second, the content of this article has been revised according to your suggestions. Thank you for your valuable comments.
Third, the charts in this article have been re-organized, hoping to show it more clearly.
Fourth, the content and format of this article have been adjusted, hoping to meet your requirements.
Fifth, according to your suggestion, some tables have been changed to bar charts. The more deviation of cellulose content may be related to the determination method. The determination of cellulose content needs to decompose the treated cellulose sample into soluble monosaccharides, which are easy to be oxidized. In the same batch of samples, the oxidation degree of each sample may different, which leads to that the cellulose content deviation seems larger in the data.
Sixth, the experimental method part in this article has been supplemented and modified.
Best wishes,
Guangzheng Li and Degang Zhao
Guizhou university
Round 2
Reviewer 1 Report
I do appreciate the improvements performed in this revised version and acknowledge the authors for the thoughtful effort. However, I still failed to see that authors improved their interpretation on the reconstructed lignin biosynthesis pathway in re-numbered figure 15. Specifically, authors must recognise that this pathway is highly conserved in plant species, and refer to the first reports of the lignin biosynthesis in poplar, a model tree species even for rice as far as this pathway is concerned. For instance, please refer to the report Tree Genet Genome 2012 8:821-829, by reinforcing the point that SNP variability (for instance in terms of figure 5’s Δ SNP index) is more likely found in downstream genes within the lignin biosynthesis pathway.
Meanwhile, I still find too many main figures for a single paper. A total of 17 individual figures is very excessive, which makes results' readability very hard. Please combine comparable figures within figure panels with several subfigures and send other redundant ones to supplemental material.
Author Response
Dear Reviewer:
I have carefully revised my article based on the your comments. Here I will response each of the point you raised.
First, the content of this article, especially chapter 2.5. Lignin metabolomics analysis, has been revised according to your suggestions. Thank you for your valuable comments.
Second, the charts in this article have been re-organized and some of them have been moved to supplemental material, hoping to show it more clearly.
Best wishes,
Guangzheng Li and Degang Zhao
Guizhou University